# Maternal Emotional Intelligence and the Provision of Child Motor Affordances

**DOI:** 10.3390/children9101442

**Published:** 2022-09-22

**Authors:** Saeed Valadi, Carl Gabbard, Saeedeh Sadat Sadrolsadati, Marzieh Elyasi

**Affiliations:** 1Department of Physical Education and Sports Science, Urmia University, Urmia 5756151818, Iran; 2Department of Health & Kinesiology, Texas A&M University, College Station, TX 77843, USA; 3Department of Physical Education and Sports Science, Arak University, Arak 3848170001, Iran

**Keywords:** emotional intelligence, affordances, home environment, motor development

## Abstract

*Background:* Mothers are not only the axis of an ecological subsystem for their children but are also capable of creating a significant emotional and psychological environment through constant and direct interaction. This leads to interactions and emotional connections with the children, supportive behaviors, and a long-term commitment to their upbringing and development. *Aims:* This study examined the relationship between maternal emotional intelligence (EI) and demographics on the availability of motor affordances in the home environment that are conducive to their child’s motor development. *Methods:* Mothers (*N* = 451) and homes of children aged 18–42 months were assessed using the Persian version of Affordances in the Home Environment for Motor Development-Self-Report (AHEMD-SR) and the Bar-On Emotional Intelligence Questionnaire. In addition, mothers’ physical activity (PA) experience and demographic data were collected. *Results:* Analyses indicated significant correlations between mothers’ education, level of PA, and income with EI and motor affordances. *Conclusions:* EI, PA experience, and select mother demographics were important to the availability of motor affordances in the home. One could also speculate, based on previous research, that such provision may affect the future motor development of the child. Future research should include a younger and more diverse population.

## 1. Introduction

Research clearly indicates that the environment plays a critical role in childhood development. That is, an optimal level of development occurs with a stimulating environment and strong contextual support; support that is rooted in the home. Whereas considerable past research supports the notion, contemporary inquiries, including reviews, demonstrate that the home environment plays a vital role in physical and mental development and well-being [1,2,3]. Contemporary research shows that the level of motor development is a significant factor in child behavior, including cognitive and social-emotional aspects [4,5,6,7].

Within the home, our interests are in the provision of affordances that promote child motor development. Founded strongly in Gibsonian theory, affordances are agents of and events in the environment that afford the individual potential for perception and action; action that results in learning and development of a skill and/or a part of the cognitive/musculoskeletal systems. Affordances include toys, apparatus, materials, availability of space, and stimulation and nurturing by parents (and others). In summary, the home environment has the potential to provide a plethora of opportunities that can be conducive to stimulating child motor development, especially at an early age [8,9].

Over the last decade, a substantial body of research points to the observation that the level of motor affordances in the home may have a negative or positive impact on child motor development; that is, higher levels of affordance provision are significantly associated with higher levels of motor development, and alternatively, lower levels are related to poorer motor behavior [10,11,12,13,14]. Higher affordance levels have also been linked to families with higher socioeconomic status and higher education [15,16], higher cognitive/communication skills [7,10], and better social development [7,17]. Much of the research noted above assessed the home using either the *Affordances in the Home Environment for Motor Development* (AHEMD) for children 18 to 42 months of age [18] or the infant version—AHEMD-IS for infants 3–18 months [19]; the former, used in the present study, is discussed in greater detail in the Methods section.

Underscoring our interests are the factors that underlie the level of affordance provision, in this case, the emotional intelligence (EI) of the mother. According to Mayer and Salovey [20], EI refers to one’s capacity to perceive, process, and regulate emotional information accurately and effectively, and most importantly, to use this information to guide one’s thinking and actions and to influence others. In our case, we are interested in a mother’s EI and how it influences the perceived needs of the child in the home. Goleman asserts that there are five key elements to EI: self-awareness, self-regulation, motivation, empathy, and social skills [21]. EI is an emotional characteristic that helps regulate emotions and impulses and increases interaction and empathy with others [22], including the mother/child dyad. Studies show that mothers with higher levels of EI are more receptive to their child’s needs, spend more time with their children, and give more importance and attention to positive parenting [23,24,25,26].

A review of research associated with the mother-child dyad clearly shows that the quality of the interactions between this dyad can both positively and negatively influence cognitive, language, motor, and social outcomes [27,28]. In closer connection to our question of home affordance and mother’s EI, Esmaeelzadehazad and colleagues found that higher EI scores were significantly associated with higher child motor skill scores [29]; motor affordance provision was not examined. The current study builds upon previous work by examining the association between maternal EI, mother’s physical activity experience (PA), select mother demographics, and the provision of motor affordances in the home; a feature that has an impact on a child’s motor development. PA was added to the list of variables due to previous reports of parental PA influence on child PA [30,31]; and more specific to our study, Mori and colleagues’ finding that parents’ self-reported PA was significantly related to higher home motor affordance scores [32]. Our initial expectation was that higher states of the variables described would be associated with greater attention to a child’s home environment in the form of motor affordance provision.

## 2. Materials and Methods

### 2.1. Participants and Procedure

This study involved 451 Iranian mothers with children aged 18 to 24 months. The average age was 27.82 ± 8.55 months (52% boys and 48% girls). They were selected using a multi-stage cluster sampling method involving selected health centers in four urban cities in Iran. Because of the COVID-19 virus presence at the time of this study, 36 centers were randomly selected from among the accessible urban health centers, of which only 19 centers accepted the invitation to participate. Participating mothers were sent, mostly via e-mails, the qualification criteria, participation materials, and consent forms provided by the health centers. After consent was received, mothers were sent weblinks to complete the EI and AHEMD questionnaires. Included with the weblinks was a brief explanation regarding how to complete the forms. This was carried out instead of a more direct face-to-face data collection due to virus restrictions limiting such exposure. For those mothers who preferred, phone calls and/or written materials were provided.

The qualification criteria stated that mothers had to be at least 16 years of age, have a high-school equivalent diploma, have a child living with the parent(s) since birth, and the child had no mental or physical impairment. This investigation was approved by the Ethics Committee of Urmia University (IR.URMIA.REC.1400.009). Mothers were assured that all information would be confidential and any results would be published without family member identification.

### 2.2. Measures

In addition to the EI and AHEMD questionnaires (described in subsequent sections), an online questionnaire accompanied the weblinks/and written materials for the collection of the mother’s age, occupation (employed or home caretaker), income, and PA experience. The AHEMD questionnaire asked questions about the child’s gender and age. PA was asked as a general question regarding the mother’s experience (past and present) with sports and/or a regular exercise routine. This was collected using a 5-option Likert scale: none (no experience history), less than one year, between 1 and 2 years, between 2 and 3 years, and more than 3 years.

#### 2.2.1. Emotional Intelligence

The *Bar-On Emotional Quotient Inventory (EQ-I)* was used to assess mothers’ EI [33]. The self-report questionnaire has 133 questions addressing 5 scales and 15 subscales: (1) intrapersonal skills (self-regard, self-assertiveness, emotional self-awareness, independence, and self-actualization), (2) interpersonal skills (empathy, social responsibility, and interpersonal relationships), (3) stress management (stress tolerance and impulse control), (4) adaptability (reality testing, flexibility, and problem-solving), and (5) general mood (optimism and happiness). The instrument is suitable for individuals aged 16 years and older and scored on a five-point Likert scale (strongly agree with ‘5’ and strongly disagree with ‘1’). Items are summed to yield a total score which reflects total EI. Bar-On tested the tool on 3831 people from six countries including Iran. The reliability of the instrument was 0.97 and its validity was confirmed by factor analysis [22]. Others have reported the EQ-I to be a valid and reliable test [34,35], including its use in Iran. In the present study, the Iranian 90-item version, with a reliability of 0.93 and validity of 0.91, was used [36].

#### 2.2.2. Motor Affordances

The *Affordance in the Home Environment for Motor Development–Self-Report (AHEMD-SR)* was used to evaluate the quality and quantity of motor affordances. The instrument was designed for children 18 to 42 months of age [18,37]. This self-report questionnaire encompasses five factors including outside space (OS), inside space (IS), variety of stimulations (VS), and fine- and gross motor toys (FMT and GMT). There were three types of questions: simple two-choice questions, 4-choice Likert questions, and descriptive questions including 20 variables and 67 items. The questionnaire has been translated into eight languages including Iranian Farsi [13]. A total score of <10, 10–15, and 16–20 is noted as representing “poor and low opportunity”, “moderate and sufficient and available opportunities”, and “high with excellent opportunities”, respectively [13,18,37]. In this study, the Persian version of the AHEMD was used with a reported validity of 0.92, test-retest reliability equal to 0.91, and internal consistency of 0.92 [13].

### 2.3. Statistical Analysis

Descriptive statistics were used to describe the data, frequency, mean, standard deviation, normality (Shapiro–Wilk test), and the parity of variances (Levene’s test). Pearson correlation tests, Spearman, and Eta were used to determine the relationships between the variables. To examine the effect of independent variables on the dependent variable, a simple and stepwise linear regression coefficient was used. Group comparisons were performed using *t*-student parametric tests and one-way variance analysis (ANOVA) with follow-up analyses using the Bonferroni statistic. To gain insight on age, child age was categorized and analyzed into four groups: 18–23, 24–29, 30–36, and 37–42 months. In addition, structural equation modeling was used to map and determine the relationships between the mothers’ EI and AHEMD variables. The significance level was set at *p* < 0.05.

## 3. Results

### 3.1. Participant Characteristics

Participants included 236 (52%) boys and 215 (48%) girls with an average age of 27.82 months (*SD* = ± 8.55) and an age range of 18 to 42 months. A weak and significant correlation was obtained between total AHEMD score and age (*p* < 0.05). Among different dimensions of the AHEMD, the only significant relationship was observed between age and Toys (FMT, GMT) (*r* = 0.20, *p* < 0.001). The significant difference between the age and toys was associated with children in age groups of 18 to 23 months and 36 to 42 months (*p* < 0.001). Children aged 36 to 42 months had more toys than children in the other age groups, especially 18 to 23 months old. There was no significant relationship between the child’s age and the mother’s EI and its dimensions.

Maternal age at the time of study ranged from 20 to 46 years (*M* = 32.37 ± 5.70). Results of the correlation between maternal age and total AHEMD showed a positive and significant relationship (*r* = 0.13, *p* < 0.01). Toys was the only dimension among other home dimensions that had a significant relationship with maternal age (*r* = 0.10, *p* < 0.01). There was no significant relationship between maternal age and the mother’s EI and its domains.

Sixty-six percent (66%, 299) of mothers were housewives, of whom 66% had a high school diploma and/or post-graduate education; of those, 70% had bachelor’s (38%) and 32% reported master’s/doctoral degrees, thirty-two percent (32%) were employed outside of the home. Statistical analysis showed that the correlation between a mother’s EI score and occupation was significant (Eta = 0.09, *p* < 0.05), and the difference between those values was significant (*t* = −1.975, *p* < 0.05). Employed mothers had higher EI than those not employed. When comparing employed and not employed mothers, of the five EI domains, two were significantly different, stress management (Eta = 0.12, *p* < 0.05) and general mood (Eta = 0.13, *p* < 0.05). That is, employed mothers had a better level of general mood skills and stress management than housewives. Furthermore, there was a significant relationship between the mother’s education level and EI (Eta = 0.14, *p* < 0.05).

Complementing those findings was a significant relationship between total AHEMD score and maternal occupation (Eta = 0.16, *p* < 0.01), with a significant difference (*t* = −3.315, *p* < 0.001). It appears that employed mothers were able to provide a richer environment of motor affordances. Furthermore, the AHEMD dimensions of Toys (Eta = 0.19, *p* < 0.001) and IS (Eta = 0.10, *p* < 0.05) showed a positive and significant correlation with mothers’ occupation. Regarding maternal education and AHEMD dimensions, there were significant differences with IS (Eta = 0.16, *p* <0.01) and Toys (Eta = 0.24, *p* < 0.001). That is, as mothers’ education increased, IS and Toy availability increased. Mothers with postgraduate/doctoral education had the highest number of toys in the home. The only significant difference between IS and education level was with the two groups of mothers with bachelor’s and master’s/doctoral degrees (*p* < 0.05); the homes of higher-educated mothers provided more inside space.

Family income, according to national standards, was divided into three groups: low (36%), average (32%), and high (32%). To no surprise, comparative differences between income and mothers’ education level indicated a significant difference (*p* < 0.01). Typically, as mothers’ education level increased so did the economic status of the family. Relatedly, there was a significant relationship between total AHEMD score and family income level (Eta = 0.16, *p* < 0.01, *F* welch = 21.929, *p* < 0.01). In other words, mothers whose family income was higher revealed higher EI scores.

Regarding participation in physical activities, 40% responded with no ‘regular’ experience, 20% reported less than 1 year, 14% between 1 and 2 years, 9% 2 and 3 years, and 17% responded as more than 3 years. Results indicated that there was a significant relationship between level of experience and total AHEMD scores (*r* = 0.29, *p* < 001). According to the results of the linear regression analyses, PA experience accounted for 27% of the total AHEMD score. Post hoc tests revealed that mothers who had 1 to more than 3 years of experience had higher AHEMD scores than mothers who had less than 1 year. Within the AHEMD dimensions, the only significant relationship was with Toys (*r* = 0.031, *p* <0.001) and the IS (*r* = 0.17, *p* < 0.001). EI results indicated significant relationships between PA experience and two domains: general mood (*r* = 0. 15, *p* < 0.001) and intrapersonal skills (*r* = 0.14, *p* < 0.001).

### 3.2. Mother’s EI and Motor Affordances

The average total maternal EI score was 327.27 ± 36.741 (Table 1), which, according to EI standards, is in the upper category. The total AHEMD total score was 13.72 ± 2.576 which indicates an average level of AHEMD.

Table 2 shows the relationships between AHEMD dimensions with EI. Results revealed that there was a significant relationship (*r* = 0.14, *p* < 0.001) between EI and total AHEMD, and a significant relationship between AHEMD and all EI domains except adaptability skills (Table 2). In addition, there were positive and significant relationships between EI and all dimensions of the AHEMD, while the OS dimension was only related to the stress management domain.

To investigate the effect of EI on AHEMD scores, linear regression analyses (Table 3) revealed that the coefficient of determination (*R^2^*) was 0.03, which indicates that 3% of the changes in EI were related to changes in the AHEMD. Furthermore, the beta coefficient for total AHEMD was equal to 0.16, which shows that the ratio of change in EI with AHEMD was equal to 0.16. This means that a unit increase in the standard deviation of EI led to an increase of 0.16 in the standard deviation of total AHEMD.

Stepwise regression analyses were used (Table 4) to determine the effect of the concurrence of the time combination of variables related to the AHEMD and mother characteristics (EI, age, occupation, education, PA experience), along with family income. The results in predicting total AHEMD identified a significant pattern (*p* < 0.001) in four stages. In the first stage, mothers’ PA experience (*R^2^* = 0.08) was included in the model as the best predictor of total AHEMD. In general, 16% (*R^2^* = 0.16) of the variance related to total AHEMD was explained only by mothers’ PA experience, income, age, and EI.

Next, to determine the relationships between the variables affecting total AHEMD, a structural model (Figure 1) was drawn with the fittings of the general model (Table 5) revealing that 14% (*R^2^* = 0.14) of the AHEMD variance was related to the variance of PA experience, family income, maternal age, and EI. These results indicated that the most impactful variables on total AHEMD were family income, PA experience, maternal age, and EI.

## 4. Discussion

This cross-sectional study was conducted to determine the effect of mothers’ EI and select mother demographics on the provision of motor affordances in the home environment; affordances that have been shown to promote child motor development. Our findings indicated that affordance provision increased with increasing maternal and child age, especially with Toys and IS (inside space). This result was in line with our expectations. An interesting note with the present data, mothers who had more than one child and completed their second child’s information for this study had greater levels of motor affordances, especially in the dimension of Toys. The reason for this could be due to the experience gained both in terms of the mother’s age and awareness of the needs of their second child.

In the present study, mother demographics played a prominent role in the presence of motor affordances. We found that income, education, and occupation (employed outside the home/home caretaker) had a significant relationship with home affordance availability. We wish to note that the examination of mothers’ occupation in this context had not been previously reported. Of course, it should be noted that even home caretakers with higher education and a good level of family income also provided ‘adequate’ affordances. Therefore, in line with previous studies, higher family income and parental education are associated with the provision of richer home affordances, especially in the dimensions of fine- and gross motor toys [16,29].

Regarding our primary variable maternal EI and mother demographics, a few findings are worthy of note. First, employed and more highly educated mothers, who also had a good SES, had higher EI scores than home caretakers with less education and lower SES levels. Speculatively, several factors could account for the difference. Employed mothers, due perhaps to greater communication and presence in social environments, have more opportunities to exchange ideas and obtain additional care information [38]; therefore, the reason could be that these mothers try more to compensate for their absence by enriching the home and play environment. Our results confirm this general observation with the finding that the number of fine- and gross motor toys and IS were better for those children whose mothers were employed outside of the home.

Another novel variable we examined, PA experience, presented some interesting insight. Mothers who reported more experience, 1 year to more than 3 years, compared to no experience, provided a greater level of motor affordances. Speculatively, mothers with more experience in PA are more aware of the benefits and need for such, for themselves and their children. This idea is in line with the studies of Sugihara and colleagues [39] and Mori and colleagues [32]. This effect of experience may also be associated with maternal EI given that in our study mothers with greater PA experience also had higher EI scores.

Total EI alone explained the total AHEMD score. Regarding the specific EI characteristics, with the exception of interpersonal skills, all other domains (intrapersonal, stress management, adaptability, general mood) had a significant correlation with total AHEMD. Previous studies have reported that lack of control and lack of care for negative emotions and the presence of stress and anxiety lead to disruptions in the quality and quantity of toy affordances [40], and subsequently with lesser growth of children’s motor development [41,42,43,44,45]. It appears that mothers with higher EI, which includes sufficient knowledge of their own emotions, can independently control their emotions in a positive manner and with more optimism to improve the quality and quantity of their child’s environment. This is in line with the results of Ewing et al. [46] in which mothers with higher EI pay more attention to their children and display good parenting skills [24].

As noted earlier, research has established that there is a significant relationship between the level of home motor affordances and a child’s motor development [12,16,47]. With the findings of the present study, we have added to that body of knowledge with the inclusion of the mother’s PA experience and level of EI.

## 5. Limitations

All research encompasses some limitations, and this study is no exception. Although we studied a relatively large sample from four urban cities, we recommend an extension to the study of more diverse groups, including rural areas. Second, our study included the age range of 18 to 42 months. A considerable body of research on motor affordances has been conducted on younger children using the AHEMD-IS (infant version for ages 3–18 months). Finally, PA experience was asked in the form of a general question. It is recommended that the qualitative level and type of PA, and even physical fitness of both parents and its relationship with EI and AHEMD be investigated in future research. Finally, because of the COVID-19 virus presence at the time of this study, all questionnaires were completed online, therefore there was no data collection monitoring.

## 6. Conclusions

The current study adds to an increasing body of evidence with our finding that maternal EI was related to the level of motor affordances in the home; the higher the EI, the greater the level of affordances. The overall effect was compounded by mothers’ PA experience, income, mother age, and education. Our findings underscore the conclusion of an extensive literature review by Yang and colleagues [48] noting that there is a positive correlation between the home parenting environment and the psychomotor development of young children. Furthermore, improving the home environment is a key factor in promoting early child development.

## Figures and Tables

**Figure 1 children-09-01442-f001:**
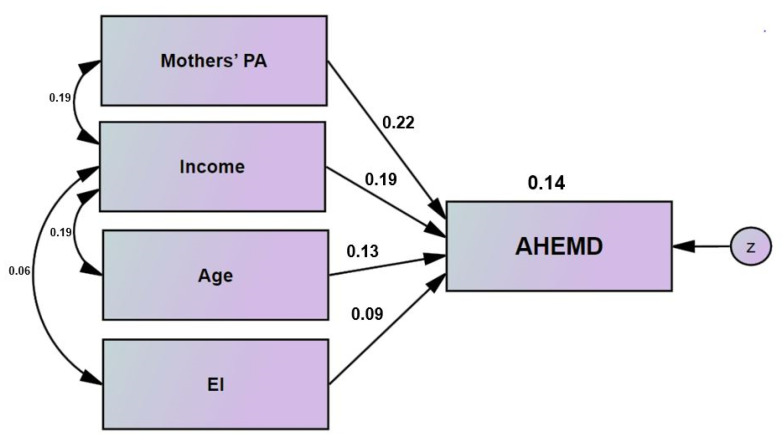
Final regression model.

**Table 1 children-09-01442-t001:** Descriptive data of AHEMD and EI.

Max.	Min.	±*SD*	Mean	
20	6	2.576	13.72	AHEMD
4	1	0.805	3.07	Outside space
4	1	0.401	3.92	Inside space
4	1	1.073	2.86	Variety of stimulation
4	1	1.000	1.96	Fine motor toys
4	1	1.087	1.90	Gross motor toys
433	217	36.741	327.27	EI
90	42	7.566	62.53	Adaptability
146	66	14.582	108.39	Intrapersonal
89	23	8.906	71.45	Interpersonal
60	14	6.738	46.14	General mood
60	15	7.66	38.75	Stress management

**Table 2 children-09-01442-t002:** Correlation between AHEMD and its factors with EI and its scales.

	EI		Stress Management		General Mood		Intrapersonal		Interpersonal		Adaptability	
	*r*	*p*	*r*	*p*	*r*	*p*	*r*	*p*	*r*	*p*	*r*	*p*
**AHEMD**	0.14 **	0.000	0.16 **	0.001	0.10 *	0.036	0.12 **	0.009	-	-	0.14 **	0.004
**Variety of stimulation**	0.12 *	0.014	-	-	0.10 *	0.041	-	-	0.15 **	0.001	0.10 *	0.035
**Inside space**	0.14 **	0.003	-	-	-	-	0.11 **	0.013	0.15 **	0.001	0.11 *	0.020
**F + GMT**	0.09 *	0.049	0.12 **	0.009	-	-	0.11 *	0.022	-	-	0.11 *	0.021
**Outside space**	-	-	0.09 *	0.050	-	-	-	-	-	-	-	-

F + GMT: Fine + Gross Motor Toys, *r*: Pearson’s correlation ratio—* *p* < 0.05, ** *p* < 0.01.

**Table 3 children-09-01442-t003:** Linear regression of EI on AHEMD and its factors.

Durbin Watson	*Sig.*	*F*	*Adj. R^2^*	*R^2^ Square*	R	*Sig.*	β-Coefficient	Coefficient B	Dependent	Independent Variable
									Variable	
2.00	0.002	11.74	0.02	0.03	0.16	0.002	0.16	0.010	AHEMD	EI
1.99	0.003	9.09	0.02	0.02	0.14	0.003	0.14	0.002	Inside space	EI
1.86	0.011	6.07	0.01	0.01	0.12	0.011	0.12	0.003	Variety of stimulation	EI
1.94	0.024	5.41	0.01	0.01	0.11	0.024	0.11	0.003	Fine motor toy	EI

**Table 4 children-09-01442-t004:** Stepwise regression of variables affecting AHEMD.

Durbin Watson	*Sig.*	*F*	*Adj. R^2^*	*R^2^ Square*	R	*Sig.*	β-Coefficient	Coefficient B	Variables	
1.90	0.001	39.47	0.08	0.08	0.29	0.001	0.29	0.448	PA	Model 1
	0.001	31.22	0.12	0.13	0.36	0.001	0.25	0.388	PA + income	Model 2
							0.21	0.597		
	0.001	24.03	0.14	0.14	0.38	0.001	0.26	0.398	PA + income + age	Model 3
							0.18	0.524		
							0.13	0.055		
	0.001	20.35	0.15	0.16	0.4	0.001	0.24	0.376	PA + income + age + EI	Model 4
							0.17	0.493		
							0.13	0.054		
							0.12	0.008		

**Table 5 children-09-01442-t005:** Model fit indices.

Parsimonious	Comparative		Absolute			Fit Indices
*AGFI*	*NFI*	*CFI*	*GFI*	*RMSEA*	*CMTN/DF*	
0/96	0/93	0/95	0/99	0/06	2/71	Confirmed
						Observed Values

*Abbreviations*: CMIN/DF = chi-square/degree of freedom, RMSEA = root mean square of error approximation, GFI = goodness-of-fit index, CFI = comparative fit index, NFI = normed fit index, AGFI = adjusted goodness of fit index.

## Data Availability

The datasets are available without personal identification, due to privacy and ethics restrictions.

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
