# Peer review of "Maternal Emotional Intelligence and the Provision of Child Motor Affordances"

_children, 2022, doi:10.3390/children9101442_

Round 1
Reviewer 1 Report
Thank you for the opportunity to review this manuacript. It is of interest for the community providing information on the relation between mother emotional intelligence and children physical activity level. It bring the attention on the importance of the home environment. The manuscript is well written and presented but require further references.
"Within the home..." please, add references for these sentences
"Over the last decades, a substantial body..." please add references
"Mother are not only the axis if..." please, add references
"Please double check format (see statistical analysis section)
End of discussion, please correct "Inappear that..."
Author Response
September 6, 2022
Dear Editor and Reviewers,
Firstly, we wish to thank you for the generally positive acceptance of the paper in Children, and second, we are very thankful for the comments that we feel have added significantly to the overall readability and clarity of our intent and findings. Please see below, for a point-by-point response to the reviewers’ comments and suggestions. The revised manuscript file with tracked changes is complete.
Reviewer #1
It is of interest for the community providing information on the relation between mother emotional intelligence and children physical activity level. It brings attention on the importance of the home environment. The manuscript is well written and presented but require further `references. (in the total manuscript, we added 7 refs).
"Within the home..." please, add references for these sentences (no need, this is an introductory sentence of general thought, no research).
"Over the last decades, a substantial body..." please add references (there were 4, but we added one found for 2020; Valentini et al. ).
"Mothers are not only the axis if..." please, add references (added).
"Please double check format (see statistical analysis section) (we checked and found no need for revision, thank you).
End of discussion, please correct "In appear that..." (done)
Reviewer #2
INTRODUCTION It is very clarifying, however, I think it is somewhat lacking. I would like to see something more about why the importance of the mother's emotional intelligence work, or research that has analyzed the effects of having a high emotional intelligence on other people (in case there are no "mothers to children”, but perhaps from coaches or teachers to students or athletes, who may be related to the study). As noted in the introduction, the literature on this topic is sparse. We have added a more descript meaning of EI and its link our study. In reference to the second comment, refs 21-26 are recent studies of mother’s EI and the child. Unfortunately, we could find none with teachers or coaches.
MATERIALS AND METHODS
Was accessibility to the centers by conviction or random? “….36 centers were randomly selected from among the accessible urban health centers,…” Likewise, it is indicated that the questionnaires have a link with an explanation to be complete. Does this mean that there was no qualified person at the time of passing the questionnaires? That is correct, unfortunately; however, this is not an unusual strategy. (if so, include in limitations, done). The number of the ethics committee should also be added. (clarifications added to text)
There is an extra space in some words, but I can't tell the exact line as it is not marked (before because for example) or in The AHEMD (measures). Be careful with the font size also before the results section. (done). Was some kind of procedure done to purify the sample? To avoid possible random responses or follow patterns? (such as outlier detection or mahalanobis distance). We used frequency tables to identify outlier data in nominal and ordinal variables and box plots to identify outlier data in distance/relative variables.
RESULTS
The participant characteristics text should be in the method. And I also think that it would be much clearer if you put a figure or a table with the data (whether it is left in the participant or if it is left here. It is true that when making the significance values, perhaps it can be left here but with a diagram flowchart or explanatory table) Done - After considering the reviewer's comment about the 'adding a table in the participant characteristics section', it is our opinion that in this section it is not warranted. We mentioned the main statistical propositions; also, we included five important/basic tables and one figure in the text, a descriptive table does not seem necessary.
The discussion and conclusion seems appropriate to the study. Thank you
References should be checked, 44 for example is not correct. (done).
Very good work
Reviewer 2 Report
Thank you very much for your interesting article, I think it can have a place in this magazine taking into account a series of considerations
INTRODUCTION It is very clarifying, however, I think it is somewhat lacking. I would like to see something more about why the importance of the mother's emotional intelligence work, or research that has analyzed the effects of having a high emotional intelligence on other people (in case there are no "mothers to children ”, but perhaps from coaches or teachers to students or athletes, who may be related to the study)
MATERIALS AND METHODS
Was accessibility to the centers by conviction or random? Likewise, it is indicated that the questionnaires have a link with an explanation to be complete. Does this mean that there was no qualified person at the time of passing the questionnaires? (if so, include in limitations). The number of the ethics committee should also be added.
There is an extra space in some words, but I can't tell the exact line as it is not marked (before because for example) or in The AHEMD (measures). Be careful with the font size also before the results section. Was some kind of procedure done to purify the sample? To avoid possible random responses or follow patterns? (such as outlier detection or mahalanobis distance)
RESULTS
The participant characteristics text should be in the method. And I also think that it would be much clearer if you put a figure or a table with the data (whether it is left in the participant or if it is left here. It is true that when making the significance values, perhaps it can be left here but with a diagram flowchart or explanatory table)
The discussion and conclusion seems appropriate to the study
References should be checked, 44 for example is not correct
Very good work
Author Response

(The authors gave the same response as above.)
